# Improved YOLOv7 Algorithm for Detecting Bone Marrow Cells

**DOI:** 10.3390/s23177640

**Published:** 2023-09-03

**Authors:** Zhizhao Cheng, Yuanyuan Li

**Affiliations:** School of Mathematics and Physics, Wuhan Institute of Technology, Wuhan 430205, China

**Keywords:** YOLOv7, bone marrow cell detection, attention mechanism, CoTLAN, focal loss

## Abstract

The detection and classification of bone marrow (BM) cells is a critical cornerstone for hematology diagnosis. However, the low accuracy caused by few BM-cell data samples, subtle difference between classes, and small target size, pathologists still need to perform thousands of manual identifications daily. To address the above issues, we propose an improved BM-cell-detection algorithm in this paper, called YOLOv7-CTA. Firstly, to enhance the model’s sensitivity to fine-grained features, we design a new module called CoTLAN in the backbone network to enable the model to perform long-term modeling between target feature information. Then, in order to cooperate with the CoTLAN module to pay more attention to the features in the area to be detected, we integrate the coordinate attention (CoordAtt) module between the CoTLAN modules to improve the model’s attention to small target features. Finally, we cluster the target boxes of the BM cell dataset based on K-means++ to generate more suitable anchor boxes, which accelerates the convergence of the improved model. In addition, in order to solve the imbalance between positive and negative samples in BM-cell pictures, we use the Focal loss function to replace the multi-class cross entropy. Experimental results demonstrate that the best mean average precision (mAP) of the proposed model reaches 88.6%, which is an improvement of 12.9%, 8.3%, and 6.7% compared with that of the Faster R-CNN model, YOLOv5l model, and YOLOv7 model, respectively. This verifies the effectiveness and superiority of the YOLOv7-CTA model in BM-cell-detection tasks.

## 1. Introduction

The examination and differentiation of BM cells’ morphology is an important means in the diagnosis for many blood and BM diseases. It is also a common laboratory procedure in clinical workflows [1,2]. Although a host of other methods are now available, such as cytogenetics [3], immunophenotyping [4], and, increasingly, molecular genetics [5], these methods are very sophisticated and highly dependent on experimental manipulation [6]. Furthermore, not all institutions have the corresponding facilities and capabilities. Therefore, cytological examination remains an important first step in the diagnostic workup of many intramedullary and extramedullary pathologies [7]. By observing and analyzing the morphological characteristics of BM cells, we are able to quantify cells of different lineages to determine the proportion of each cell, which helps in the classification of many benign and malignant hematological disorders [8]. However, BM aspirate differential cell counts (DCCs) have been difficult to automate, which is why microscopic examination and classification of single-cell morphology is still largely performed by human experts in routine clinical workflows.

Manual analysis of DCCs currently performed in clinical laboratories is suboptimal mainly due to several factors [9]. Firstly, manual assessment of BM smears requires labor-intensive visual inspection of the smears using a microscope. This technique can cause damage to the examiner’s eyesight, shoulders, and neck. Secondly, the different areas of observation selected by different examiners, as well as the understanding of the test criteria, are not completely consistent, and the test results can be subject to subjectivity [10]. Thirdly, the relatively few cells typically counted inherently results in statistical imprecision. Certainly, conventional automated hematology analyzers that do not utilize digital images have been explored for performing DCCs. However, this method has some limitations, including failure to count nucleated red blood cells and to differentiate stages of cell development, as well as interference by BM lipids [11,12]. If one utilizes the digital images feature of BM smears to develop an automatic DCC system, then these problems can be solved [13].

Over the past few years, cell detection and classification has actually become the most widely studied problem in computational pathology. Commercial hematology analyzers have begun using automated image analysis of Wright-stained smears [14]. However, achieving accurate results heavily depends on the precise control of variables, such as reducing staining variations and cell crowding, while simultaneously maximizing the retention of cytology details. Cell- and nucleus-detection algorithms often rely on regular shapes and may fail to correctly detect cells with irregular or multilobed nuclei. Accurate cell detection and localization of cell boundaries are crucial for correct classification, especially in the BM, where subtle differences in cytologic features can complicate the process.

With the rapid development of deep learning, object-detection algorithms are gradually applied to cell detection. However, most previous studies on automated cytomorphological detection and classification have focused on leukocyte types of peripheral blood smears or physiological cell types [15,16,17,18,19,20], limiting their usefulness for diagnostic leukocyte detection of hematological malignancies in the BM. Blood cells in peripheral blood smears are easier to recognize, as they contain only five types of WBCs: segmented neutrophils, eosinophils, basophils, lymphocytes, and monocytes. Compared with peripheral blood smears, BM smears have a higher density of contacting and overlapping cells, more cell types, and more complex cell morphologies, so the object-detection task is more challenging. The current literature on BM aspirate smear image analysis does not have high-speed and efficient cell-detection methods, and has been successful in only a few cytological categories, limiting its potential clinical application, which is a particularly challenging problem [21,22].

Object-detection models can be mainly divided into two-stage models and one-stage models. When the model size is the same, the two-stage models usually exhibit higher classification accuracy, but their time cost is much higher than that of the one-stage models. Among the two-stage models, there is a very classic Faster RCNN proposed by S. Ren et al. [23]. In the one-stage object-detection models, You Only Look Once (YOLO) is a representative model, such as YOLOv3, proposed by J. Redmon et al. [24], and YOLOv5, which is very popular in the industry. However, compared with the above algorithm, the YOLOv7 model published by Wang et al. [25] on CVPR in 2023 has faster speed and higher accuracy on the COCO dataset. In academia, YOLOv7 is a more advanced, more accurate, and faster object-detection algorithm that can meet the needs of different application scenarios. It supports both mobile GPUs and GPU devices from edge devices to the cloud. Therefore, it can be combined with an electron microscope scanner and applied to Wise Information Technology of med (WITMED) to quickly and effectively complete the automation task of BM aspirate DDCs. However, there are very few studies on the application of the YOLOv7 model in BM cell detection. At the same time, the accuracy of the model in detecting BM cells also has room for improvement, because the accuracy of this detection method is susceptible to subtle differences in myeloid cell types, small cell size, and dense cell distribution.

In response to the above problems, this paper proposes an improved BM cell detection algorithm, YOLOv7-CTA, for the detection of bone marrow cells in BM aspire smears. This manuscript provides the following contributions: In order to improve the sensitivity of the model to the fine-grained features of BM cells, we hope that the model can perform long-term modeling of the target feature information in the picture and improve the model’s ability to extract fine-grained features. Therefore, a new feature-extraction network module, CoTLAN, is designed in this paper to solve this problem.To alleviate the impact of image complex background on object detection, this paper introduces the self-attention mechanism CoordAtt module and combines it with the CoTLAN module.Through comparative experiments, this paper selects the most suitable loss function CIoU as the localization loss function of the network.To address the slow convergence issue caused by the small size of BM cells, this paper uses the K-means++ algorithm to cluster the target boxes in the BM cell dataset to generate more suitable anchor boxes. In addition, this paper uses Focal loss function to replace the original multi-classification cross entropy to solve the imbalance between positive and negative samples.

The experimental results show that the improved YOLOv7-CTA has a better detection effect on 15 types of BM cells contained in BM aspirate smears than others.

## 2. Materials and Methods

### 2.1. Dataset Preparation and Preprocessing

#### 2.1.1. Image Acquisition

Since public datasets of BM cells are scarce and involve too few cell types, we sought the help of a medical testing institution. To this end, we spent a lot of human and material resources obtaining real BM aspirate smear samples. With the help of experts, we objectively labeled the collected cell images to sort out a real and valid dataset. The specific workflow for producing the BM cell image dataset is illustrated in Figure 1. 

The process of producing a BM cells dataset can be broadly divided into four steps: Firstly, a small amount of BM aspirate from patients with hematological disorders is obtained, and the smear is made after Wright staining. Secondly, the smears are scanned using an electron microscope, resulting in an image with a resolution of 88,070 × 89,011 pixels. Then, an image acquisition application with the highest resolution for collecting BM-cell images, a resolution of 600 × 600, is developed by analyzing the image file, and a preview function is provided to facilitate the collection process. Finally, with the assistance of doctors and experts, the cell images are annotated. 

#### 2.1.2. Image Database and Label Database

In this experiment, the ground-truth boxes were labeled on the cell image by using the Labelme tool [26]. Because the number of cells such as macrophages and plasma cells are too few, they were not involved in this experiment. We selected 1204 BM-cell images, a total of 13,059 BM cells, and these cells contained a total of 15 categories. Figure 2 shows these 15 types of BM cells in our database, each of which includes various shapes.

The dataset of labeled BM cells was partitioned into training and validation sets, with a 7:3 split ratio. Here, we counted the number of different types of cells and plotted the size distribution of cell labels; the results are shown in Figure 3.

In Figure 3a, the vertical axis is the number of labels, and the horizontal axis is the label name. It can be seen from the figure that there are relatively large differences in the number of different types of cell targets to be detected, but this distribution is more in line with the actual situation. In Figure 3b, the width of the abscissa represents the ratio of the label width to the image width, while the height of the ordinate signifies the ratio of the label height to the image height. The target size of this dataset is relatively small; its width and height are about 9% of the image size.

### 2.2. Proposed Methods

The entire detection framework proposed in this paper is shown in Figure 4, which is divided into four parts. 

First of all, this paper changes the network structure of YOLOv7. Due to the small size of BM cells, the feature difference between cells is not obvious. In order to capture the fine-grained features of BM cells effectively, we design a novel feature-extraction network module called CoTLAN in this paper. This module facilitates the long-term modeling of target feature information, thereby enhancing the model’s sensitivity to the subtle characteristics of BM cells. At the same time, in order to cooperate with the CoTLAN module’s attention to small target features, this paper introduces the self-attention mechanism CoordAtt module to reduce the impact of irrelevant features on object detection. Secondly, this paper replaces the initial anchor boxes. The K-means++ algorithm is utilized to cluster all ground-truth boxes of BM-cell labels in the dataset and generate anchor boxes of varying sizes. This ensures that the initial anchor box size of the model aligns with the target size, thereby accelerating the model convergence. Thirdly, this paper replaces the loss function. Focal loss is used as a classification loss in the model to solve the imbalance between positive and negative samples [27]. Finally, this paper chooses to use CIoU loss as the localization loss in the model, and adopts the CIoU non-maximum suppression (NMS) method to avoid multiple detections of the target and make the model output more accurate. The improved model improves the convergence speed and detection performance.

### 2.3. Backbone Network

The inspection model, namely YOLOv7-CTA, for BM cells in this paper is shown in Figure 5. The whole network model structure comprises four parts: input, backbone network, neck network, and head network.

In order to increase the robustness of the model, this paper also adopts the mosaic data augmentation method [28] at the input side. The idea is to stitch four defect images with operations such as random scaling, random cropping, and random layout to generate more new data samples.

The backbone network consists of CBS, MP, CoordAtt, and CoTLAN modules. The CBS module consists of regular convolution, batch normalization, and activation functions. The MP module consists of maxpooling and CBS modules. The CoTLAN module is a new module designed in this paper, which is very similar to ELAN in the YOLOv7 model, which retains the advantages of the ELAN module. Specifically, the last 3×3 convolutional module of a standard ELAN module is replaced by a CoTBlock module, which can aggregate dynamic context information and local convolutional static context information, effectively improving the model’s expressive ability to fine-grained features. The size of the feature map is not changed after the CoTLAN module, but the number of output channels at the end of the module has changed. In addition, the CoordAtt modules are added after the MP modules of the backbone, which helps the model to more accurately locate and identify the target of interest.

The neck network consists of a feature pyramid network (FPN) and path aggregation network (PAN). The backbone network’s feature map output undergoes 32-fold downsampling and is then processed by the SPPCSP module, which reduces the channel count from 1024 to 512. Subsequently, the feature map undergoes feature fusion using both top-down and bottom-up approaches. The neck network structure effectively combines feature maps at various levels. Finally, the RepC and Conv modules in the head are utilized to output the prediction results.

#### 2.3.1. CoT-Block Module

In recent years, Dosovitskiy A et al. [29] borrowed and improved the Transformer structure based on self-attention to apply it to computer vision tasks and achieved good results. However, the traditional self-attention mechanism can only be based on the interaction between the query and the key, and the rich contextual connection between adjacent keys is not explored, thus limiting the feature expression ability of the self-attention mechanism. Therefore, this paper uses a novel Transformer-style self-attention module, named Contextual Transformer block [30], referred to as CoT-Block. This module is used to explore rich contextual connections between adjacent keys, while being able to capture long-range dependencies at different locations globally. Its detailed structural diagram is shown in Figure 6.

As can be seen from Figure 6, for the input feature X∈ℝH×W×C (H: height, W: width, C: channel number), the CoT-Block module first defines three variables, Q=X, K=X, and V=XWv (Wv is an embedding matrix that is implemented as a 1 × 1 convolution in space). Firstly, 3×3 group convolution operation is performed on K to obtain the feature K1∈ℝH×W×C with local static context modeling information. The above process is shown in Formula (1).
(1)K1=Conv3×3 gruop(K)

Secondly, conditioned on the concatenation of K1 and Q, two consecutive 1×1 convolution operations are performed to generate the attention matrix A∈ℝH×W×(3×3×Ch) (Ch denotes the head number for each head, and the local attention matrix at each spatial location of A is learned based on the query feature and the contextualized key feature), as shown in Formula (2), where δ is a non-linear activation function.
(2)A=Conv1×1(δ(Conv1×1([K1,Q])))

The resulting attention matrices A and V are then subjected to local matrix multiplication to obtain the global dynamic context modeling information feature K2∈ℝH×W×C, as shown in Formula (3), where ⊛ denotes local matrix multiplication.
(3)K2=A⊛V

Finally, the result of fusing the feature $K^{1}$ with local static context modeling information and the feature $K^{2}$ with global dynamic context modeling information is taken as the output result of CoT-Block.

Therefore, inspired by the above research, this paper uses CoT-Block to replace the last 3×3 convolution in the ELAN structure in the original YOLOv7 model, and designs a new CoTLAN structure to enhance the feature extraction of the feature-extraction ability network of the original YOLOv7 model. The self-attention mechanism module in CoTLAN makes each pixel in the current feature map pay more attention to the features of other pixels in the current feature map, and explores the rich contextual connections between adjacent pixels. Learning the connection between different pixels through the weights between pixels can help the model extract more effective feature information from the feature map. In the feature-extraction network, CoTLAN replaces ELAN, which can effectively improve the feature-extraction ability of the model for small targets, especially in the detection scenario of BM cell images.

#### 2.3.2. CoordAtt Module

Due to the complex and changeable environment of BM cells, in order to improve the feature expression ability of the model on BM cells, an attention mechanism module was added between every two CoTLAN modules of the backbone network.

Attention mechanisms can be categorized into channel attention mechanisms, spatial attention mechanisms, and a combination of both. While traditional attention modules like Squeeze-and-Excitation attention (SE) [31] and Convolutional Block Attention Module (CBAM) [32] excel at modeling channel-to-channel relationships, they tend to overlook spatial positional information. On the other hand, other attention modules that address this issue have excessive parameters, which hinder their deployment in real-world applications. 

However, CoordAtt mechanism [33] also uses directional perception and positional perception in addition to capturing cross-channel information, thus facilitating precise localization and identification of the targeted objects by the model. Additionally, the CoordAtt module is versatile and can be added at multiple locations in existing models. The framework diagram of the CoordAtt attention mechanism is shown in Figure 7.

As can be seen from Figure 7, in order to avoid all spatial information being compressed into channels, the global average pooling of the input is decomposed here. Specifically, given an input X, we encode each channel along the horizontal and vertical coordinates, respectively, using two spatial extents, (H,1) or (1,W), of the pooling kernel. Therefore, the output of the c-th channel at height h and the output of the c-th channel with width w are expressed as Formula (4) and Formula (5), respectively:(4)$$z_{c}^{h}\left(h\right)=\frac{1}{W}\underset{0\leq i<W}{\sum }x_{c}\left(h,i\right)$$
(5)$$z_{c}^{w}\left(w\right)=\frac{1}{H}\underset{0\leq j<H}{\sum }x_{c}\left(j,w\right)$$

Secondly, the feature map zcw(w)∈ℝ1×W×C is transposed and spliced with the feature map zcw(w)∈ℝH×1×C, and then the 1 × 1 convolution operation with activation is performed to generate the feature map f∈ℝ1×(H+W)×C/r, as shown in Formula (6):(6)f=δ(Conv1×1([zch,zcw]))

Along the spatial dimension, f is split into fh∈ℝ1×H×C/r and fw∈ℝ1×W×C/r. Then, 1 × 1 convolution is used to increase the dimension and is then combined with the sigmoid activation function to obtain the final attention vectors gh∈ℝ1×H×C and gw∈ℝ1×W×C, as shown in Formulas (7) and (8):(7)gh=δ(Conv1×1(fh))
(8)gw=δ(Conv1×1(fw))

Finally, the outputs $g^{h}$ and $g^{w}$ are, respectively, expanded and used as attention weights, and the output of the coordinate attention block Y can be written as Formula (9):(9)$$y_{c}\left(i,j\right)=x_{c}\left(h,i\right)\times g_{c}^{h}\left(i\right)\times g_{c}^{w}\left(j\right)$$

The input undergoes horizontal and vertical pooling to retain long-range dependencies in both directions. The resulting information is then concatenated and split, followed by convolutions that emphasize both horizontal and vertical directions. The resultant two-part feature map can be precisely localized to the target object’s row and column of interest. 

### 2.4. Anchor Box Optimization

In order to speed up the detection accuracy and efficiency of the improved model, this paper uses the K-means++ [34] algorithm to cluster the target boxes in the BM-cell dataset to generate more suitable anchor boxes. The K-means++ algorithm selects one cluster center at a time, which not only allows the random center-point to approach the local optimal solution, but also approximates the global optimal solution.

Specific steps are as follows:
(1)Randomly select a BM-cell sample target box as the initial cluster center, and then calculate the minimum intersection ratio distance D(x) between this center and the rest of the sample boxes.(2)Compute the probability *P*(*x*) of each BM-cell sample box being selected as the next cluster center, and employ the roulette method to determine the next cluster center.(3)Iteratively perform steps (1) and (2) until all cluster centers have been selected.(4)Compute the distance between each BM cell sample and the cluster centers. Allocate each sample to the class associated with the cluster center with the smallest distance, and update the cluster center $c_{i}$ of each category. Continue updating the classification and cluster center iteratively until size remains constant.

The formula for the classification loss function is as follows:(10)$$D\left(x\right)=1-IoU_{\left(x,c\right)}$$
(11)$$P\left(x\right)=\frac{D{\left(x\right)}^{2}}{\sum_{x_{i}\in X}D{\left(x_{i}\right)}^{2}}$$
(12)$$c_{i}=\frac{1}{\left|c_{i}\right|}\underset{x\in c_{i}}{\sum }x$$
where *x* represents the sub-target mark sample box, c denotes the center of the cluster, IoU denotes the intersection-over-union ratio between two rectangular boxes, X denotes the total sample of the target marker frame, *i* = 1, …, *k*, in this paper, the detection model consists of three detection feature maps, and each feature map corresponds to three anchor boxes, resulting in k=9, $\left|c_{i}\right|$ denoting the number of samples belonging to the cluster center as $c_{i}$ category. Table 1 shows the optimized anchor boxes corresponding to the three feature map sizes.

The large-sized anchor box corresponds to the 20×20 feature map and is responsible for detecting large objects in the image. The medium-sized anchor box corresponds to the 40×40 feature map and is responsible for detecting medium-sized objects. The small-sized anchor box corresponds to the 80×80 feature map and is responsible for detecting small objects.

### 2.5. Loss Function

The loss function of the model comprises three components: localization loss (box_loss), confidence loss (obj_loss), and classification loss (cls_loss). The total loss is the weighted sum of the three losses. Among them, the confidence loss use binary cross-entropy loss.
(13)$$Loss=w_{1}\times box\_loss+w_{2}\times cls\_loss+w_{3}\times obj\_loss$$
where $w_{1}$, $w_{2}$, and $w_{3}$ are the weight values of the three loss functions, respectively.

In a given image sample, the BM cells correspond to the foreground, while the remaining regions are regarded as the background. Nevertheless, the YOLO series, as one-stage object detectors, suffer from sample complexity and class imbalance issues caused by a disproportionate number of positive and negative samples [27]. These problems can adversely affect the network’s gradient update direction and lead to lower accuracy. To address the aforementioned issues, an improved version of the cross-entropy function is proposed, namely the Focal loss function, which is specifically designed for addressing the class imbalance problem in object detection. By assigning different weights to the loss function based on the sample difficulty level, the Focal loss function can assign less weight to easily distinguishable samples and greater importance to complex distinguishable samples.

The formula for the classification loss function is as follows:(14)$$cls\_loss=-\alpha {\;}_{t}{\left(1-p_{t}\right)}^{\gamma }\log\left(p_{t}\right)$$
(15)$$p_{t}=\{\begin{array}{c} p\;\;\;\;\;\;\;if\;y=1\\ 1-p\;otherwise \end{array}$$
(16)$$\alpha {\;}_{t}=\{\begin{array}{c} \alpha \;\;\;\;\;\;\;if\;y=1\\ 1-\alpha \;\;otherwise \end{array}$$
where y=1 denotes the real sample and p is the probability that the sample predicted by the model belongs to the foreground. To solve the imbalance of sample categories, the weight parameter α is introduced. The adjustment factor γ is added to the cross-entropy loss function.

### 2.6. Non-Maximum Suppression

This paper conducts comparative experiments on the selection of the current mainstream IoU loss function, and finally chooses CIoU loss function as the localization loss function of the model. At the same time, this paper uses the CIoU-NMS [35,36] algorithm to filter the preliminary prediction box output by the image to be recognized. The final prediction box can be obtained with the following steps:(1)Set both the confidence threshold and the CIoU threshold.(2)Compute the confidence level of all prediction boxes output by the network model. Sort the prediction boxes that exceed the confidence threshold in descending order of confidence form high to low, and put them into the candidate list.(3)Select the prediction box with the highest confidence from the candidate list, save it to the output list, and remove it from the candidate list.(4)Compute the CIoU loss between the initial prediction box with the highest confidence obtained in the previous step and all other preliminary prediction boxes in the candidate list. Remove the prediction box from the candidate list if its cross-merger loss is higher than the CIoU threshold.(5)Iteratively perform steps (3) and (4) until the candidate list is depleted.(6)Use the prediction boxes in the output list as the final.

## 3. Results

### 3.1. Experimental Environment and Evaluation Metrics

To validate the efficacy of our model, we conducted neural network training and testing using the computer configuration parameters and hyperparameter settings in Table 2.

The evaluation indicators of the model mainly included precision and recall, whose expressions are as follows:(17)$$Precision=\frac{TP}{TP+FP}$$
(18)$$Recall=\frac{TP}{TP+FN}$$
where TP corresponds to true positives where the model prediction is positive and the actual value is also positive; FN represents false negatives where the model prediction is negative, but the actual value is positive; FP indicates false positives where the model prediction is positive, but the actual value is negative; and TN denotes true negatives where the model prediction is negative, and the actual value is also negative. 

The accuracy of the model was assessed using the mean average precision (mAP), which considers both precision and recall. 

### 3.2. Experimental Results and Analysis

Firstly, we conducted comparative experiments and ablation experiments to improve the model network structure using the YOLOv7 model as the baseline. The comparative experiment analyzed three of the most popular attention mechanisms to identify the optimal one, while the ablation experiment aimed to validate the effectiveness of the proposed model-structure improvement method. Secondly, we used the improved YOLOv7-CTA model as a baseline to determine a suitable localization loss function, and conducted ablation experiments on the setting of anchor box hyperparameters and the use of the improved loss function to further optimize the model. Finally, we compared the improved model’s performance with the original model and other classic models and verified its superiority. Since the main purpose of this paper is to improve the detection performance of the model, we used metrics such as precision, recall, mAP, and FPS (Frames Per Second) to evaluate the results. 

To incorporate suitable attention mechanisms into the network, this paper added SE, CBAM, and CoordAtt attention mechanisms between ELAN modules for training and comparison. The results of the experiments are presented in Table 3. 

The impact of adding different attention mechanisms to the backbone of model detection varies. While the SE attention mechanism ignores positional information, it considers channel attention. The result shows that adding SE attention to the backbone improves the mAP, but it improves the least. The CBAM attention aims to leverage positional information by reducing the input tensor’s channel dimension and computing spatial attention through convolutional operations. However, convolution can only capture local relationships and cannot model the long-term dependencies required for visual tasks. The incorporation of CBAM attention into the backbone improves the mAP, but it significantly increases the number of parameter counts and computational costs, and decreases by 4 FPS. In contrast, the CoordAtt attention mechanism embeds positional information into channel attention, capturing long-range dependencies in one spatial direction while preserving accurate positional information in the other. The results show that after adding CoordAtt to the backbone network, although the recall is reduced by 1.2%, the precision and the most important evaluation index, mAP, increase by 8.0% and 3.6%, respectively, which are significantly higher than the previous two methods, and the increase in parameters and calculation is not significant. This indicates that using the CoordAtt attention mechanism can enable the network model to detect targets more extensively and improve the network’s detection ability. 

Ablation experiments were conducted in this study to validate the beneficial effects of the improved strategy proposed on the network. The results are presented in Table 4, where the adoption of the corresponding enhancement method is denoted by “√” and the absence of the enhancement method is denoted by “×”. 

The detection outcomes of the original YOLOv7 network are presented in the first row of the table. As can be seen from the above table, when the ELAN module of the original model is replaced with the new designed structure, CoTLAN, compared with the original YOLOv7 without any improvement strategy, precision, recall, and mAP are improved by 6.4%, 0.4%, and 3.6%, respectively. The network model integrates the two modules of CoTLAN and CoordAtt. The results show that the recall rate and FPS of the combined method have decreased, but the accuracy and mAP have been greatly improved. Among them, the detection accuracy and mAP increased by 14.9% and 5.2%, respectively, which effectively improved the detection ability of the model.

After the network model was determined, we used the improved YOLOv7-CTA model as the baseline and conducted comparative experiments on different localization loss functions. We compared the performances of three new methods: SIoU [37], EIoU [38], and CIoU (ours). The results are shown in Table 5.

As seen from Table 5, when the CIoU loss function is used, the accuracy value and mAP value of the improved model are the highest. From this, we conclude that using the CIoU loss function can help the improved model to have a better detection effect on BM cells.

This paper finally uses CIoU loss as the localization loss function of the YOLOv7-CTA model, and conducts ablation experiments on the setting of anchor box hyperparameters and the use of the improved loss function, as shown in Table 6.

After replacing the original initial anchor box with the new anchor box generated by the K-means++ algorithm, the mAP of the improved model increased by 0.4%. This proves the effectiveness of the method. When the cross-entropy loss function in the improved model is replaced by the Focal loss function, the improved model improves mAP by 0.6%. For the combination of these two improvements, the mAP is significantly improved to 88.6%, which surpasses the previous two methods. For the above improvement schemes, this paper selects the improvement scheme with the highest mAP to participate in the subsequent experimental comparison.

Since the PR curve comprehensively considers the accuracy and recall of each category detected by the model, in order to better measure the performance of the model, we further used the PR curve to measure the performance of the model.

Figure 8 depicts the PR curve of the present model, where the horizontal axis indicates the recall rate, and the vertical axis represents the precision. The PR curve provides a visualization of how precision changes as the recall rate increases. If the area enclosed by the curve and the coordinate axis in the figure is larger, it means the overall performance of the model is better. Compared with the original model, the performance improvement of the model is very obvious. It can be seen from the figure that the mAP of N-segmented and basophil decreased slightly, and the mAP of other cells increased to varying degrees. Among them, the mAP of seven types of cells reached more than 92.5%, indicating the effectiveness of the YOLOv7-CTA model.

While the ablation experiments confirmed the effectiveness of the improved strategy proposed in this study relative to the original algorithm, it remains to be established whether it can achieve an advanced level. As such, a series of comparative experiments were conducted on BM-cell datasets, under identical experimental conditions, to compare the proposed method with the current mainstream object-detection approach. Figure 9 illustrates the comparison of training outcomes among multiple models. The results demonstrate that the proposed algorithm in this paper achieved notably higher precision and mAP than the other two models.

The training loss curves of various models are compared in Figure 10. The curves of all models stabilized and converged after 250 iterations. The results indicate that YOLOv5l exhibited a significantly inferior performance to YOLOv7 concerning both regression and classification losses. From the figure, the improved network structure in this paper resulted in a slightly slower convergence rate of the model compared with that of YOLOv7. Nonetheless, after approximately 70 iterations, the proposed model exhibited a superior decline rate and convergence ability to YOLOv7, suggesting the efficacy of the loss function adjustment in enhancing the network’s convergence performance.

Finally, this paper lists the comparison results of the performance metrics of different models, as shown in Table 7. The table shows that, compared with other models, the recall, precision, and mAP of YOLOv7-CTA are significantly higher. Among them, compared with YOLOv7, the precision, recall, and mAP of the improved model have increased by 10.1%, 0.6%, and 6.7%, respectively. Although the processing speed of YOLOv7-CTA is slightly lower than that of YOLOv7, it is still much faster than the two-stage model Faster R-CNN. The FPS indicator shows that YOLOv7-CTA can recognize 22 images of BM cells with a resolution of 600 × 600 per second, so the expected running time for the detection and recognition of 500 BM cells is <2.5 s, which can help pathologists greatly reduce their workload. In conclusion, the improved model has achieved high speed and high efficiency in predicting the task of BM cells. 

In addition, we calculated the parameter count and computational cost of these models. Although the parameters of the two-stage model are small, the model structure is very complicated because the detection and classification are performed separately, resulting in a particularly large amount of calculations and high training costs. However, our proposed YOLOv7-CTA has the lowest computational cost, which reduces 2.7GFLOPs on the basis of YOLOv7, and its parameter count is also reduced by 0.6 M. Therefore, the YOLOV7-CTA does not have high requirements for hardware equipment, and has lower requirements than other models. 

## 4. Discussion

In order to more intuitively verify the actual detection effect of the model after improving the feature-extraction network, this paper selects three images from the test set of the BM-cell dataset for detection. The detection effect is shown in Figure 11, where the red arrow points to the case of false detection. The improved model, YOLOv7-CTA, has significantly more outstanding detection ability than other models. 

Limited research has been conducted on automating BM aspirate DCCs using image analysis. Choi et al. [21] presented preliminary outcomes utilizing convolutional networks for cell classification in DCCs. However, their dataset consisted solely of 2174 non-neoplastic erythroid and myeloid precursor cells, excluding other crucial cell types like eosinophils, basophils, monocytes, lymphocytes, and plasma cells. Their study primarily focused on classification and did not consider detection, relying solely on manually cropped cell images for classifier development and validation. Reta et al. [22] devised a comprehensive software pipeline for cell detection and classification to identify acute leukemia subtypes. Their dataset involved 633 cells from acute lymphoblastic leukemias and acute myeloid leukemias. They implemented advanced image processing techniques to detect and segment leukocytes in digital images of Wright-stained BMA smears. The segmented cells were then characterized using various features describing their shape, color, and texture. Subsequently, basic machine learning algorithms were employed to classify individual cells, and these cell classifications were aggregated to provide a single diagnosis for the sample. Although their approach is limited in scope, targeting a few specific cell types, they reported high-precision segmentation accuracy. Wu et al. [39] proposed the BMSNet model based on the YOLOv3 for BM puncture fluid-cell recognition, which is also a one-stage model. The study showed that while the model performed well with a small number of cells and few classes, they evaluated the model’s predictive performance on bounding boxes, finding that the average precision without considering the classification was 67.4%. This compares poorly with our mAP performance for identifying all cells in the validation set. 

Our model builds upon the groundwork laid by these earlier publications, highlighting the potential and promise of deep learning methods in automating DCCs. However, the improvement of the model is still insufficient. The improvements reported in this work are primarily attributed to modifications made to the network backbone structure, while the detection head also plays a crucial role in feature fusion and model optimization. In order to more comprehensively optimize our network model, we will conduct further research in the future. Compared with the original model, the improved model has a slower inference speed. We can consider using knowledge distillation to compress the model to speed up the inference speed of the model. In addition, we performed relatively little in terms of data enhancement, because we have not found more enhancement methods suitable for the bone-marrow-cell scene, and we hope that more researchers will conduct further research on this. 

## 5. Conclusions

In this paper, an improved BM-cell detection algorithm, YOLOv7-CTA, is proposed. This method can identify BM-cell images more accurately than similar models. Experimental analysis shows that a new feature-extraction network, CoTLAN, is designed in the backbone network, which can improve the extraction capability of fine-grained features. The CoordAtt module is combined in the network to make the model pay more attention to the features in the area to be detected, reducing irrelevant features, thereby enhancing the feature-extraction ability for small targets. Finally, under the determined network structure, the model is optimized through the selection of the localization loss function, the use of the K-means++ algorithm for clustering the target frame of the BM-cell dataset, and the replacement of cross entropy. The experimental results show that the mAP of the optimized model reaches 88.6%, surpassing the Faster R-CNN, YOLOv5l, and YOLOv7 models by 12.9%, 8.3%, and 6.7%, respectively. Furthermore, the detection speed of this model is 22 FPS, effectively satisfying the requirements for high performance. Compared with other models, the YOLOv7-CTA model has superiority in the BM-cell-detection task.

There are still some improvements to be made. At present, in the BM puncture smear image dataset, the number of some species is rare, and some species are not even included in the dataset. In the image data, the cell imaging effect is average, the microscope scanner equipment is relatively common, and the cytoplasm and nucleus features of some stained cells are not obvious. Therefore, we can improve the dataset and obtain a BM puncture smear image dataset with more comprehensive cell data, clearer cell imaging, and more uniform cell positions, so as to further research the detection of BM cells. In addition, under certain standards, our model will also cause recognition errors when the differences between cell types are extremely subtle, which also shows that there is still room for improvement in the model. At the same time, how to reduce the differences between labeling standards is a problem that needs to be solved. Importantly, we mainly focused on improving the model, and did not conduct auxiliary diagnosis experiments for specific BM or blood-related diseases. We hope that, in the future, there will be opportunities to quantify the effectiveness of the model for medical diagnosis, making it a valuable addition to the medical field. 

## Figures and Tables

**Figure 1 sensors-23-07640-f001:**
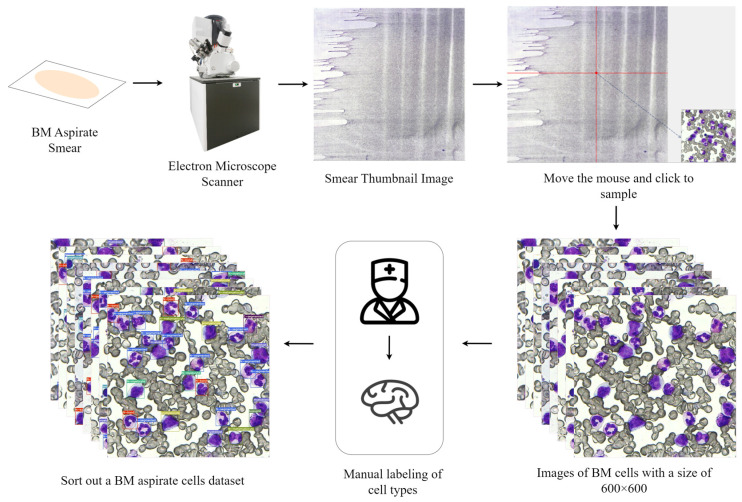
The specific process of making a BM aspirate cells dataset.

**Figure 2 sensors-23-07640-f002:**
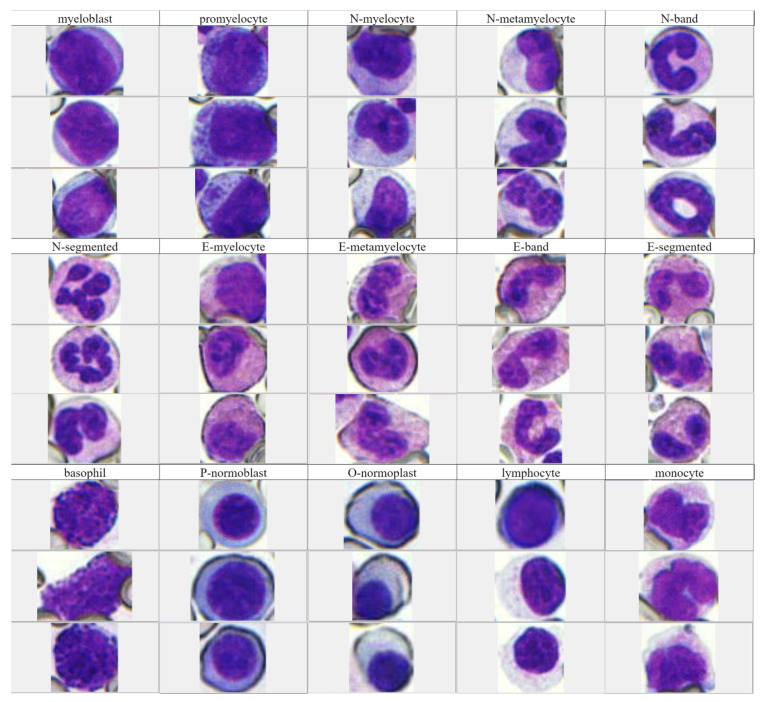
There are images of 15 types of cells collected from BM aspirate. (N: neutrophil, E: eosinophilic, P: polychromatic, O: orthochromatic).

**Figure 3 sensors-23-07640-f003:**
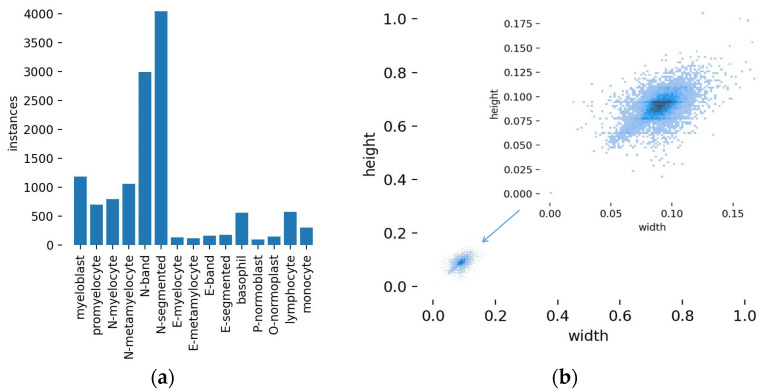
(**a**) Number of BM cell labels, (**b**) label size distribution.

**Figure 4 sensors-23-07640-f004:**
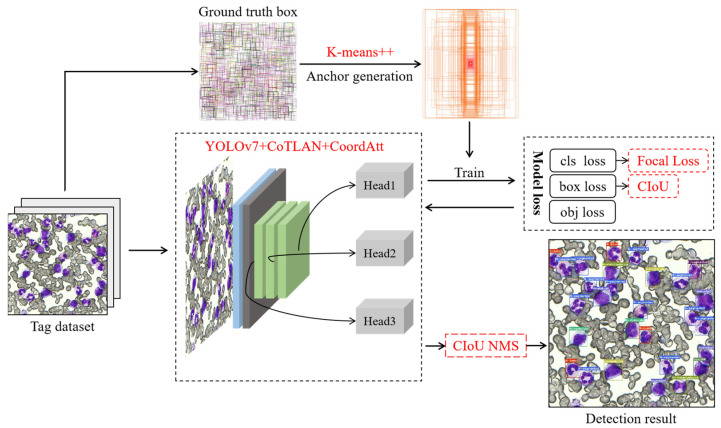
Detection framework of the proposed method.

**Figure 5 sensors-23-07640-f005:**
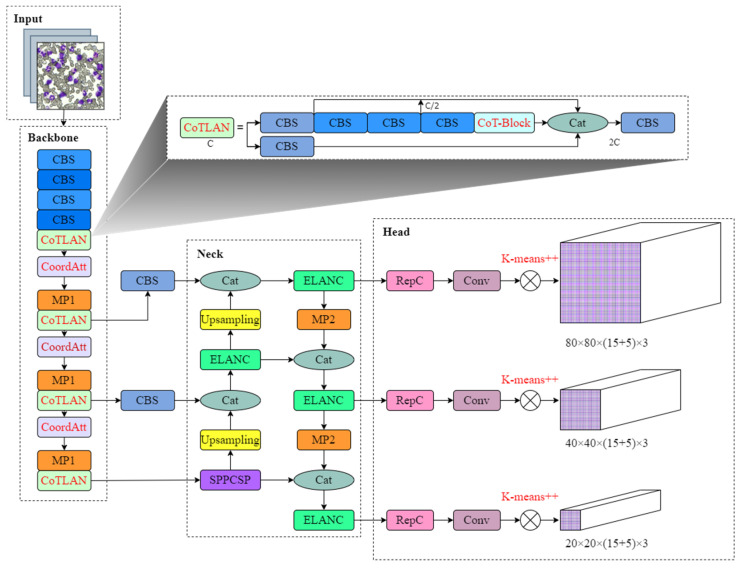
Structure of the YOLOv7-CTA network.

**Figure 6 sensors-23-07640-f006:**
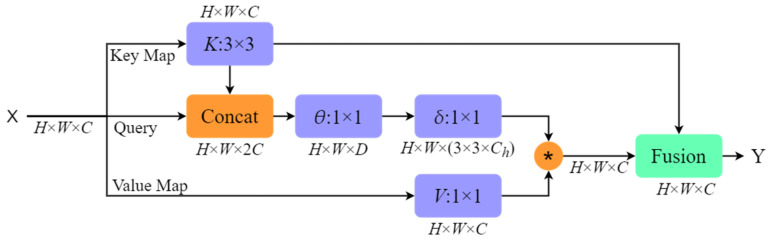
CoT-Block module structure diagram.

**Figure 7 sensors-23-07640-f007:**
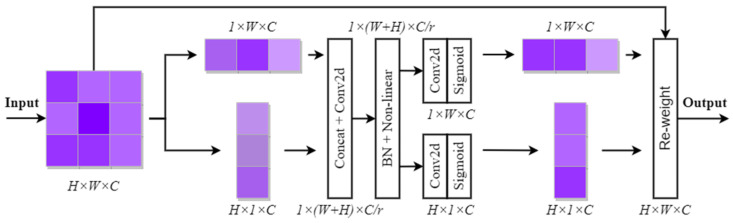
The schematic diagram of CoordAtt.

**Figure 8 sensors-23-07640-f008:**
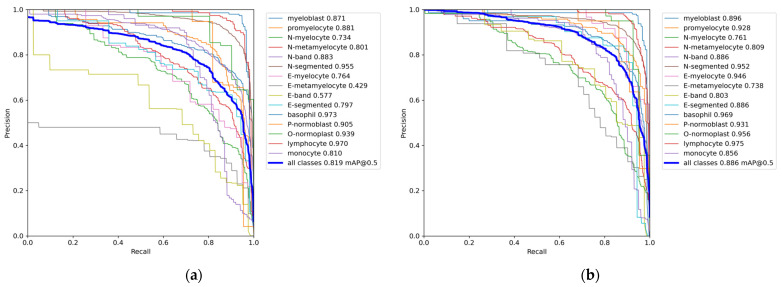
Precision–recall curve of different models: (**a**) YOLOv7 model, (**b**) YOLOv7-CTA.

**Figure 9 sensors-23-07640-f009:**
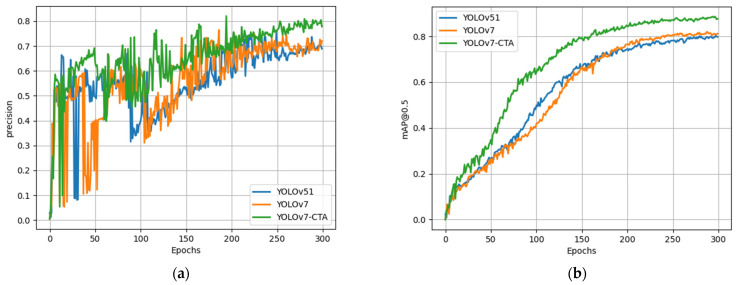
The training results of various models: (**a**) precision curve, (**b**) mAP@0.5 curve.

**Figure 10 sensors-23-07640-f010:**
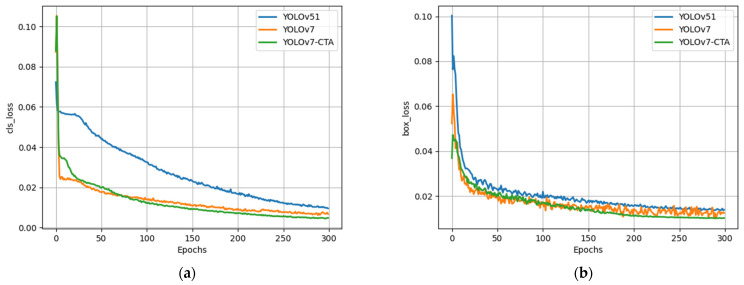
The training loss curves of various models: (**a**) cls_loss curve, (**b**) box_loss curve.

**Figure 11 sensors-23-07640-f011:**
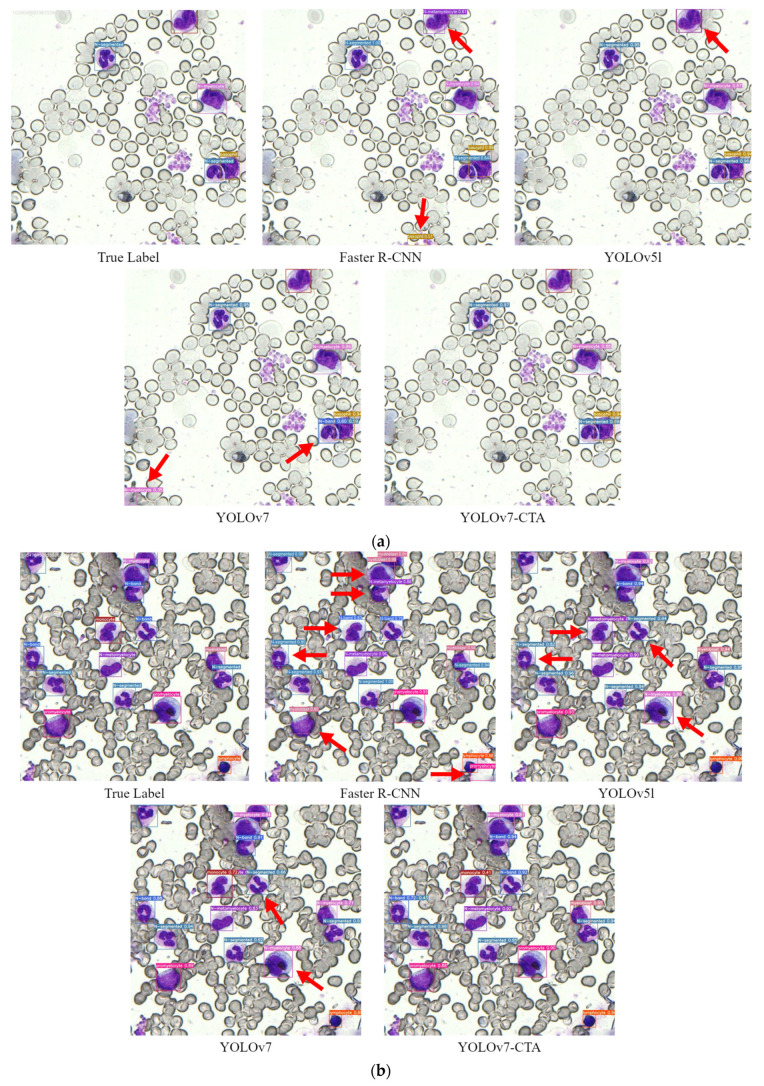
(**a**) Since we use boxes to label cells, the surrounding feature information of the cells in the boxes is also be learned. When the cell environment is complex, other models tend to treat the background as cells, but our method can avoid such errors. (**b**) This is the recognition effect of denser cells, and our model has better classification accuracy than other models. (**c**) Our model also has recognition errors, but the error rate is lower than that of other models. The wrong three-cell categories are very similar to the correct ones, which is related to the standard of labeling. Subtle differences in the standards for cell-class labeling have been problematic. In addition, there is still room for improvement in the model, and we will conduct further research in the future.

**Table 1 sensors-23-07640-t001:** Optimized anchor box parameters.

Feature Map Size	20 × 20	40 × 40	80 × 80
YOLOv7-CTA	(61, 55)	(54, 56)	(38, 39)
	(67, 64)	(57, 64)	(42, 57)
	(74, 75)	(60, 40)	(51, 49)

**Table 2 sensors-23-07640-t002:** Experimental environment configuration.

Parameter	Configuration
CPU	Intel(R) Core(TM) i7-6700CPU @ 3.40 GHz
GPU	NVIDIA GeForce GTX1660 Ti (6G)
Operating System	Windows10
CUDA	11.6
Python	3.8.5
Torch	1.12.1
Momentum	0.937
Weight decay	0.0005
Batch size	4

**Table 3 sensors-23-07640-t003:** Performance comparison of models equipped with various attention mechanisms.

Attention Mechanisms	GFLOPs	Params	Precision/%	Recall/%	mAP@0.5/%	FPS
None	103.4	36.6 M	68.3	84.5	81.9	26
SE	103.6	36.7 M	68.2	84.9	83.5	26
CBAM	113.9	38.1 M	72.6	85.1	83.8	22
CoordAtt (ours)	103.8	36.7 M	76.3	83.3	85.5	25

**Table 4 sensors-23-07640-t004:** Ablation experiment results.

Models	CoTLAN	CoordAtt	Precision/%	Recall/%	mAP@0.5/%	FPS
YOLOv7	×	×	68.3	84.5	81.9	26
	√	×	74.7	85.1	85.5	23
	×	√	76.3	83.3	85.5	25
	√	√	83.2	80.4	87.1	22

**Table 5 sensors-23-07640-t005:** Performance comparison of models equipped with various localization loss functions.

Loss	Precision/%	Recall/%	mAP@0.5/%
EIoU	66.3	84.3	79.2
SIoU	76.5	84.5	86.4
CIoU (ours)	83.2	80.4	87.1

**Table 6 sensors-23-07640-t006:** Ablation experiment results.

Models	K-Means++	Focal CIoU	Precision/%	Recall/%	mAP@0.5/%
YOLOv7-CTA	×	×	83.2	80.4	87.1
	√	×	80.4	82.6	87.5
	×	√	80.0	84.6	87.7
	√	√	78.4	85.1	88.6

**Table 7 sensors-23-07640-t007:** Comparison of performance metrics of various models.

Models	Backbone	GFLOPs	Params	Precision/%	Recall/%	mAP@0.5/%	FPS
Faster R-CNN	ResNet50	507.6	28.4 M	68.3	74.9	75.7	7
YOLOv5l	CSPDarkNet53	114.2	46.7 M	70.2	83.6	80.3	25
YOLOv7	CBS + ELAN	103.4	36.6 M	68.3	84.5	81.9	26
YOLOv7-CTA	CBS + CoTLAN + CoordAtt	100.7	36.0 M	78.4	85.1	88.6	22

## Data Availability

The code link: https://github.com/Gameness-czz/YOLOv7-CTA, accessed on 27 July 2023. The BM dataset link: https://doi.org/10.6084/m9.figshare.23805324.v1, accessed on 30 July 2023.

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
