# Peer review of "Improved YOLOv7 Algorithm for Detecting Bone Marrow Cells"

_sensors, 2023, doi:10.3390/s23177640_

Round 1

Reviewer 1 Report

This manuscript presents an improved YOLO-v7 method to detect bone marrow cell. May few suggestions about this manuscript are listed below.

1. On page 2, contributions shown in last paragraph are just messed up. Authors are advised to write contributions in bullets form.

2. Proposed method is well written. However, I suggest to authors to include a pseudo code of their developed method. This will help readers to quickly understand the algorithm.

3. Table 5 is nice and presents a good ablation study.

4. Results shown Figure 11 are great. I appreciate authors for doing such a good work.

5. Authors should also include a dedicated section about Discussion. There, please mention the strengths and limitations of the proposed method.

6. I would also suggest a dedicated section to include a computational complexity of proposed method. Or if you can show the test time to yield the final output image.

7. Number of references and quality is fine. It covers all duration from 2009 to 2023.

8. Overall research gap is addressed nicely. Hopefully, it will be valuable addition in the research domain.

Author Response

Reviewer 1: This manuscript presents an improved YOLO-v7 method to detect bone marrow cell. May few suggestions about this manuscript are listed below.

  1. On page 2, contributions shown in last paragraph are just messed up. Authors are advised to write contributions in bullets form.

Response: Thank you very much for your careful review. We are sorry that we did not clearly express the contribution of this paper. We have modified our contributions in the revised manuscript. (line 97-112, page 2)

  1. Proposed method is well written. However, I suggest to authors to include a pseudo code of their developed method. This will help readers to quickly understand the algorithm.

Response: Thank you very much for your useful suggestion. We are sorry that the description of developed method is not easy to understand. We provide the algorithm code address in the Data Availability Statement section (line 566, page19). We have modified description of developed method in the revised manuscript. (line 203-282, page 7)

  1. Table 5 is nice and presents a good ablation study.

Response: Thank you for your approval.

  1. Results shown Figure 11 are great. I appreciate authors for doing such a good work.

Response: Thank you for your evaluation.

  1. Authors should also include a dedicated section about Discussion. There, please mention the strengths and limitations of the proposed method.

Response: Thank you very much for your useful suggestion. Based on your suggestion, we added a picture of the model detecting errors in the discussion section, and mentioned where the model might still need improvement in the last paragraph. At the same time, the limitations of the model and the problems to be solved are added in the conclusion, and the work that may need to be done in the future is mentioned. We have modified our discussion and conclusion in the revised manuscript. (line 528-539 and 563-570, page 18)

  1. I would also suggest a dedicated section to include a computational complexity of proposed method. Or if you can show the test time to yield the final output image.

Response: Thank you very much for your useful suggestion. In Table 8, we have used the evaluation index FPS, which represents the number of pictures that the model can detect per second. Later, we added the two evaluation indicators of model calculation amount and parameter amount, and added specific instructions. We have modified this part in the revised manuscript. (line 477-490, page 15)

  1. Number of references and quality is fine. It covers all duration from 2009 to 2023.

Response: Thank you for your approval.

  1. Overall research gap is addressed nicely. Hopefully, it will be valuable addition in the research domain.

Response: Thank you for your affirmation.

Reviewer 2 Report

The authors present an improved YOLOv7 network, YOLOV7-CTA, designed for Bone Marrow Cell detection and identification. The network integrates the attention mechanism and improves the detection accuracy of BM cells. Comments and suggestions for this article are as follows:

 1. In the fifth paragraph of Section 1, the first sentence is not properly expressed. The first stage target detection method and the second stage target detection method are two different classifications, so the expression from the second stage to the first stage is not appropriate, and the general precision of the second stage target detection method is higher.

 2. In Section 2.2.2, this document was not cited when using Focal loss function.

 3. In Section 2.3.2, the introduction of the CoordAtt Module lacks the necessary formula description.

 4. The Param and FLOPs of the model are missing in Table 8.

Author Response

Reviewer 2: The authors present an improved YOLOv7 network, YOLOV7-CTA, designed for Bone Marrow Cell detection and identification. The network integrates the attention mechanism and improves the detection accuracy of BM cells. Comments and suggestions for this article are as follows:

  1. In the fifth paragraph of Section 1, the first sentence is not properly expressed. The first stage target detection method and the second stage target detection method are two different classifications, so the expression from the second stage to the first stage is not appropriate, and the general precision of the second stage target detection method is higher.

Response: Thank you very much for your careful review. We are sorry for our vague expressions. The one-stage models are developing rapidly, not only much faster than the two-stage model, but also the mAP of some one-stage models is higher than that of some two-stage models. However, when the model size is consistent, the classification accuracy of the two-stage model is usually higher. We have modified inappropriate expression in the revised manuscript. (line 77-80, page 2)

  1. In Section 2.2.2, this document was not cited when using Focal loss function.

Response: Thank you very much for your careful review. We have put the reference in Section 2.5. (line 324, page 10). We have modified cite order in the revised manuscript.(line 170, page 5)

  1. In Section 2.3.2, the introduction of the CoordAtt Module lacks the necessary formula description.

Response: Thank you very much for your useful suggestion. We have modified this part in the revised manuscript. (line 263-277, page 9)

  1. The Param and FLOPs of the model are missing in Table 8.

Response: Thank you very much for your careful review. Following your suggestion, we have added two evaluation indicators, Params and GFLOPs, and evaluated and analyzed their values. In summary, our method has minimal requirements on hardware devices. We have modified this part in the revised manuscript. (line 477-490, page 15)

Reviewer 3 Report

1. improve the literature in the introduction chapter.

2. Make sure the introduction clearly conveys the importance of accurate bone marrow (BM) cell detection and classification. Provide more context on why current methods fall short and why this proposed YOLOv7-CTA algorithm is needed. Highlight the significance of the problem in hematology diagnosis and its real-world impact. This will help readers understand the motivation behind your work.

3. While you've outlined the main components of the YOLOv7-CTA algorithm, consider providing more detailed explanations of the CoTLAN module and the Coordinate Attention (CoordAtt) module. Explain how these modules work in improving the detection accuracy and what specific problems they address. Additionally, illustrate the integration of these modules within the network architecture with diagrams or visual aids for better clarity.

4. Provide a thorough description of your experimental setup, including details about the dataset, training configuration, hyperparameters, and evaluation metrics used. In the results section, discuss not only the quantitative results (mAP improvement) but also qualitative aspects. Include visual examples of successful detections and compare them with other models' results. This will give readers a better understanding of the model's performance.

5. Emphasize the potential impact of your proposed YOLOv7-CTA model on the workload of pathologists. Discuss how the increased accuracy and efficiency of the algorithm can alleviate the burden of manual identification. Quantify the potential reduction in manual identification efforts and highlight the implications for improving overall diagnostic accuracy and patient care.

6. Acknowledge any limitations of your proposed approach. For instance, address potential scenarios in which the model might still struggle or fail, and explain how these could be mitigated in the future. Additionally, suggest avenues for further research and improvement. This could include exploring other types of data augmentation techniques, integrating domain-specific knowledge, or investigating the potential use of more advanced neural network architectures.

Moderate editing of English language required

Author Response

  1. improve the literature in the introduction chapter.

Response: Thank you very much for your careful review. We have added some necessary documentation. We have modified this part in the revised manuscript. (line 30-33, page 1)

  1. Make sure the introduction clearly conveys the importance of accurate bone marrow (BM) cell detection and classification. Provide more context on why current methods fall short and why this proposed YOLOv7-CTA algorithm is needed. Highlight the significance of the problem in hematology diagnosis and its real-world impact. This will help readers understand the motivation behind your work.

Response: Thank you very much for your insightful comments. In the introduction, we introduced some of the difficulties encountered by current DDCs, and described some of the current methods used for medical diagnosis and their limitations, and explained some of the difficulties in the bone marrow cell data set, thus leading to why it is necessary to propose YOLOv7- CTA algorithm. Based on your suggestion, we detailed the limitations and advantages of some current research methods in the Discussion section. We have modified our discussion in the revised manuscript. (line 506-530, page 17)

  1. While you've outlined the main components of the YOLOv7-CTA algorithm, consider providing more detailed explanations of the CoTLAN module and the Coordinate Attention (CoordAtt) module. Explain how these modules work in improving the detection accuracy and what specific problems they address. Additionally, illustrate the integration of these modules within the network architecture with diagrams or visual aids for better clarity.

Response: Thank you very much for your useful suggestion and careful review. We are sorry that the description of developed method is not easy to understand. In the revised manuscript., we have used the algorithm framework diagram and added specific formulas and descriptions to help readers understand the algorithm more easily. (line 203-282, page 7)

  1. Provide a thorough description of your experimental setup, including details about the dataset, training configuration, hyperparameters, and evaluation metrics used. In the results section, discuss not only the quantitative results (mAP improvement) but also qualitative aspects. Include visual examples of successful detections and compare them with other models' results. This will give readers a better understanding of the model's performance.

Response: Thank you very much for your useful suggestion. Following your suggestion, we have added formula descriptions of the necessary evaluation metrics in the Results section. Then, We have added example plots in Figure 11, compared the detection performance of our model and other models, and explained the possible reasons for the results in the successful and wrong examples respectively. We have modified these parts in the revised manuscript. (line 361-370, page 11. And line 498-505, page 17 )

  1. Emphasize the potential impact of your proposed YOLOv7-CTA model on the workload of pathologists. Discuss how the increased accuracy and efficiency of the algorithm can alleviate the burden of manual identification. Quantify the potential reduction in manual identification efforts and highlight the implications for improving overall diagnostic accuracy and patient care.

Response: Thank you very much for your useful suggestion and insightful comments. In Table 8, we have used the evaluation index FPS, which represents the number of pictures that the model can detect per second. Later, we estimated that the model's time to detect and identify 500 bone marrow cells is <2.5 seconds, thereby quantifying the potential reduction in manual identification effort. Overall, our method has better performance. We have modified this part in the revised manuscript. (line 478-491, page 15)

  1. Acknowledge any limitations of your proposed approach. For instance, address potential scenarios in which the model might still struggle or fail, and explain how these could be mitigated in the future. Additionally, suggest avenues for further research and improvement. This could include exploring other types of data augmentation techniques, integrating domain-specific knowledge, or investigating the potential use of more advanced neural network architectures.

Response: Thank you very much for your useful suggestion.Based on your suggestion, we add the limitations of the model and the problems to be solved in the conclusion, and mention the work that may need to be done in the future. And added in the discussion where the model may still need to be improved. We have modified our discussion and conclusion in the revised manuscript. (line 528-539 and 563-570, page 18 )

Reviewer 4 Report

The paper in well written. I have some comments:

-In order to understand, the performance of the proposed method, I suggest  I suggest to add an image similar to image 11 in which the proposed method failed. From this image it is necessary to include a paragraph that describes the possible causes and the conditions that generate the lower performance.

-In the conclusions it is necessary to add some paragraph referring to the limitations of the method and under what conditions the recognition could fail or have the worst performance.

Author Response

The paper in well written. I have some comments:

  1. In order to understand, the performance of the proposed method, I suggest to add an image similar to image 11 in which the proposed method failed. From this image it is necessary to include a paragraph that describes the possible causes and the conditions that generate the lower performance.

Response: Thank you very much for your useful suggestion. Following your suggestion, we have added the (c) plot to Figure 11. In this figure, our method has three identification errors, but the error rate is the lowest compared to other models. We illustrate the reasons for the errors and the limitations of model improvement from different perspectives. We have modified this part in the revised manuscript. (line 498-505, page 17)

  1. In the conclusions it is necessary to add some paragraph referring to the limitations of the method and under what conditions the recognition could fail or have the worst performance.

Response: Thank you very much for your useful suggestion. Based on your suggestion, we add the limitations of the model and the problems to be solved in the conclusion, and mention the work that may need to be done in the future. And added in the discussion where the model may still need to be improved. We have modified our discussion and conclusion in the revised manuscript. (line 528-539 and 563-570, page 18 )

Round 2

Reviewer 3 Report

Accepted , all comments are answered. 

Moderate editing of English language required